# Shear Bond Strength of Veneered Zirconia Repaired Using Various Methods and Adhesive Systems: A Comparative Study

**DOI:** 10.3390/polym13060910

**Published:** 2021-03-16

**Authors:** Syed Rashid Habib, Salwa Bajunaid, Abdulrahman Almansour, Abdulkarim AbuHaimed, Muqrin Nasser Almuqrin, Abdullah Alhadlaq, Muhammad Sohail Zafar

**Affiliations:** 1Department of Prosthetic Dental Sciences, College of Dentistry, King Saud University, Riyadh 11545, Saudi Arabia; sbajunaid@ksu.edu.sa (S.B.); Abdulrhman.a.m.m@gmail.com (A.A.); Abdulkarimabuhaimed@gmail.com (A.A.); meqren.14@gmail.com (M.N.A.); 78abdullah78@gmail.com (A.A.); 2Department of Restorative Dentistry, College of Dentistry, Taibah University, Al Madinah, Al Munawwarah 41311, Saudi Arabia; MZAFAR@taibahu.edu.sa; 3Department of Dental Materials, Islamic International Dental College, Riphah International University, Islamabad 44000, Pakistan

**Keywords:** composite repair, crown repair, repair of zirconia, zirconia fracture

## Abstract

The aim of the present study was to investigate the shear bond strength of five different repair methods and adhesive systems for zirconia (Zr) cores layered with feldspathic porcelain. Seventy-five Zr specimens (10 × 10 × 4 mm^3^) were prepared, sintered, layered with 2 × 10 × 10 mm^3^ of feldspathic porcelain, and fired. The ceramic was fractured, and the load recorded using a shear-bond test. Specimens were thermocycled and randomly divided into 5 groups (n = 15/group) based on the repair methods. Composite repair blocks with similar dimensions to the layered ceramic (2 × 10 × 10 mm^3^) were built according to each repair method. Shear bond strength testing of the specimens with composite built up was carried out using a universal testing machine (Instron^®^5960, Massachusetts, USA). The shear bond strengths of the adhesive interface between repaired composite and the Zr were recorded for all the test groups. The fractured specimens’ surfaces were examined under a scanning electron microscope (Jeol, Musashino, Akishima, Tokyo, Japan) for evaluation of the type of failure and surface characteristics. Shear bond strength of the veneered ceramic bonded to the Zr for all the test groups was non-significant (ANOVA, *p* = 0.062). Shear bond strength after the repair revealed significant differences (ANOVA, *p* = 0.002). Group-C (13.79 ± 1.32) and Group-D (9.77 ± 4.77) showed the highest and lowest shear bond strength values, respectively. Paired Sample T-tests showed significantly lower values (*p* = 0.000) for the repaired (composite) Zr compared to the layered (ceramic) Zr. Multiple comparisons revealed differences (significant) between the shear bond strength of Group-D with Groups A (*p* = 0.010) and C (*p* = 0.003, Post Hoc Tukey test). The repair methods tested showed variations in their respective shear bond strengths. Complete ceramic/zirconia repair systems showed better bonding between the repaired composite and Zr core. The mean shear bond strength for the repaired fractured layered Zr showed acceptable outcomes in terms of clinical perspective, but was, however, unpredictable.

## 1. Introduction

A variety of materials are available for the indirect restoration of endodontically treated or badly damaged teeth e.g., all-metal, metal ceramic, all-ceramic, zirconia (Zr) veneered with ceramic, and monolithic Zr [1,2,3]. To overcome the inherent deficiencies of color mismatch of metal crowns and brittleness of all-ceramic crowns, Zr crowns were introduced for the indirect restoration of teeth with compromised coronal tooth structure [4]. Zr crowns have been used successfully for the past two to three decades due to their color, excellent biocompatibility, and mechanical properties [5]. Despite the recent introduction of more translucent monolithic systems, Zr is still more opaque than traditional all-ceramics. Therefore, the esthetics of the Zr core is further enhanced by the application of porcelain over the Zr to overcome the inherent deficiency of lack of translucency in Zr [6]. However, the bonding between the Zr core and the veneering ceramic layer makes these veneered Zr restorations weaker due to failure of adhesion between the two chemically different materials [7].

Numerous research studies have evaluated the performance of Zr crowns veneered with ceramic in function and have shown a high success rate after 2 to 5 years [8,9,10]. Nevertheless, the weakest point of these veneered Zr is the interface between the Zr core and the veneering ceramic, which in some cases has resulted in fracture/chipping of the veneering ceramic, resulting in an exposed Zr core. This bonding failure between the Zr core and veneering ceramic is the most common complication of Zr-based restorations [11,12]. Chipping of the layering ceramic does not necessarily mean a failure of the crown. However, this may become a dental emergency if it occurs in the esthetic zone of the mouth, or in the posterior region compromising the function and sometimes causing injury to the tongue or musculature [13]. In such clinical scenarios, the replacement of the complete restoration may not be the best practical solution [14]. The replacement of the restoration is time-consuming, costly, as well as having a greater risk of damage to the prepared tooth once an attempt is made for removing the damaged restoration [15].

The conventional approach to the fractured porcelain in Zr crowns is to replace the restoration rather than trying to repair it. This approach of replacement may not be the optimal choice of treatment as it poses the risk of trauma to the underlying tooth structure which can be assumed to be already compromised before the crowning [16]. Furthermore, the Zr crowns are usually cemented with resin or resin-modified glass ionomer cement which has the property of bonding chemically to the tooth structure [17]. In addition, the process of removal of the Zr core will inevitably result in damaging the underlying abutment tooth. Thus the replacement of the restorations in these scenarios is critical from the point of view of the risk to the tooth and also the added cost of fabricating a completely new restoration. Intraoral repairing of the Zr crowns is a viable solution in cases where there is partial damage to the restoration. Repairing the fractured porcelain intraorally is relatively convenient and a much cheaper and time-effective alternative to the patient and treating dentist, with the adequate restoration of function and esthetics [18].

The modes of failure of porcelain chipping or fracture have been reported in many research studies and have resulted in improved bonding of the porcelain and Zr core [19,20,21,22,23]. However, only a few research studies are available, which have discussed or suggested how to overcome the clinical situation of porcelain fracture in veneered Zr restorations [15]. 

In clinical practice, crown failure due to porcelain chip off from the Zr core usually occurs under complex types of stresses. A range of intraoral repair systems are available and tried to optimize the fractured restoration for improving the bond strength [24,25]. However, most studies are based on the clinical significance of these repair systems, to the bond strength between composite resins and fractured porcelain and/or exposed metal surfaces in ceramic-metal crowns [26,27]. The studies reporting the bond strength of intraoral repair methods for adhering composite resin to zirconia are scarce [28]. Therefore, the aim of this in vitro study was to compare the shear bond strength of five repair methods for Zr cores layered with feldspathic porcelain. The null hypothesis was that all five repair methods tested will have the same degree of shear bond strength after the repair.

## 2. Materials and Methods

The ethical approval for this in vitro study was obtained from the ethics committee at the College of Dentistry Research Center (CDRC), King Saud University (Registration # IR 0321). The study was conducted from September 2019 to January 2020.

### 2.1. Zirconia Block Preparation

Seventy-five Zr specimens (10 × 10 × 4 mm^3^) were cut with a precision saw (Isomet-2000^®^; Precision Saw, Buehler, Chicago, IL, USA) from prefabricated blocks of Zr (ZirCAD^®^; Ivoclar, Germany). The specimens were then sintered (Ceramill-Therm^®^; Amanngirrbach, Koblach, Austria) following the manufacturer’s instructions. 

### 2.2. Veneering Porcelain Application

The Zr specimens after sintering were then layered with 2 mm of feldspathic porcelain (Porcelain IPS Classic^®^; Ivoclar Vivadent AG, Bendererstrasse 2, 9494 Schaan, Principality of Liechtenstein) and fired (Programat^®^ EP3000; Ivoclar, Schaan, Germany) according to the manufacturer’s instructions. The specimens were then embedded in autopolymerising resin (Ortho-Resin^®^; DeguDent GmbH, Hanau, Germany), to facilitate the mounting of the specimen into the universal testing machine (Figure 1). The specimens were randomly divided into five groups of fifteen specimens each, based on the repair methods used for the fractured ceramic layer (Table 1).

### 2.3. Shear Bond Strength Testing of The Specimens With Ceramic Built Up

All the specimens embedded in the resin were secured and seated in a shear bond testing jig of the universal testing machine (Instron^®^5960, Norwood, MA, USA). The forces were applied at a right angle to the ceramic built up and the shear bond strength values in megapascals (MPa) were determined at a crosshead speed of 1.0 mm/min until the failure (Figure 2).

### 2.4. Thermocycling of Specimens

To simulate the clinical scenario, all the specimens were stored for 24 h at 37 °C in distilled water followed by thermocycling (Huber, SD Mechatronik Thermocycler, Germany) (5 °C and 55 °C; 6000 cycles) with a 30 s dwell and 5 s transfer time. 

### 2.5. Application of the Porcelain Repair System

The test surfaces of all the group specimens were treated following the manufacturer’s instructions provided in the manual of the materials (Table 2). 

For all the test group specimens, composite repair blocks were built with dimensions similar to the layered ceramic thickness of 2 × 10 × 10 mm^3^. To ensure the even thickness of a 2 mm composite layer for all the specimens, a silicone putty index (3M ESPE Express^TM^ STD, Maplewood, MN, USA) was used for the application of composite over the treated surfaces of the test specimens. All the surfaces of the composite built up were cured (40 s for each surface) using a light-curing unit (blue phase NMC, Ivoclar Vivadent, Germany). Prior to shear bond testing, all the 75 specimens of 5 groups were then again thermocycled in water (5 °C and 55 °C; 6000 cycles) with a 30 s dwell and 5 s transfer time.

### 2.6. Shear Bond Strength Testing of the Specimens with Composite Built Up 

Each repaired test specimen along with the resin block was locked in the metal holder of the universal testing machine (Instron^®^5960, Norwood, MA, USA). Loading at a right angle and exactly the same location used earlier for ceramic fracture and with similar speed to the repaired composite built up of the specimens was applied until failure occurred. The shear bond strengths of the adhesive interface between repaired composite and the Zr were recorded in MPa. 

### 2.7. Scanning Electron Microscopic Examination

The specimens were also analyzed for fractography using the scanning electron microscope (SEM; JEOL, JSM-6360LV, 3-1-2 Musashino, Akishima, Tokyo 196-8558, JAPAN) and evaluated for the type of failure and surface characteristics (Figure 3 and Figure 4). Each specimen was placed on an aluminum stub, gold-coated using a sputter coater (Fine coat ion sputter, JFC-1100, JEOL Ltd., Tokyo, Japan), and the surfaces to be examined were kept parallel to the base of the SEM. Specimens were examined and images were recorded by an experienced SEM technician at 100× magnification.

### 2.8. Data Analysis

All the data were computed, and mean and standard deviations of the shear bond strength were calculated for all the five test groups and subjected to analysis of variance (one-way ANOVA) and a 5% post hoc Tukey’s test for multiple comparisons between the groups, using SPSS^®^ (Ver. 22.0, SPSS, Chicago, IL, USA). The significance level was set at *p* < 0.05.

## 3. Results

The shear bond strength of the veneering ceramic over the layered Zr before and after repair with five repair methods was investigated. Shear bond strength of veneered ceramic bonded to the Zr (pre-repair), all the test groups showed similar values and were non-significant with ANOVA (*p* = 0.062). With Group-C (27.66 ± 2.35 MPa) and Group-A (30.06 ± 1.27 MPa) exhibiting the lowest and highest shear bond strength, respectively (Table 2). For the shear bond strength after the repair of all the test groups, one-way ANOVA (*p* = 0.002) revealed significant differences between the test groups. This indicated significant differences in the shear bond strength of the various tested repair systems/methods after the ceramic repair (Table 3). Among the tested repair systems, Group-C (13.79 ± 1.32) and Group-D (9.77 ± 4.77) exhibited the highest and lowest shear bond strength values, respectively (Table 3).

Comparison of the shear bond strength before (Zr core layered with ceramic) and after (repaired fractured ceramic) with the tested repair methods, for the layered Zr with Paired Sample T-test, showed that the post-repair shear bond strength for all repair systems (*p* = 0.000) was significantly lower compared to the shear bond strength values between the layered ceramic and the Zr core (Table 3). According to the results, the average shear bond strength of the repair methods for the fractured ceramic Zr specimens was less than half (42.51%) of the original bond strength between the layered ceramic and Zr core. The highest mean difference for the shear bond strength between the pre and post ceramic repair was observed for Group-D (18.98) and the least for Group-C (13.86) (Table 4).

Table 4 describes the one-to-one comparison between the five tested repair methods for the shear bond strength using the Post Hoc Tukey HSD test that exhibited significant differences while comparing the shear bond strength of Group-D with Groups A (*p* = 0.010) and C (*p* = 0.003). While the comparison between post-repair shear bond strength values between the rest of the groups was statistically non-significant (*p* ≥ 0.05) (Table 5). 

## 4. Discussion

In the current in vitro research study, the shear bond strength of the veneering ceramic over the layered Zr before and after repair with five different repair methods was investigated. For this purpose, Zr specimens with feldspathic porcelain facing were tested for shear bond strength, subjected to thermocycling treatment, and repaired using different systems. The repaired specimens were analyzed for the shear bond strength and type of fracture mode using SEM. Achieving a stronger interface of two materials is an important parameter in terms of the clinical success of any restoration [29]. Therefore, the reliability and durability of these chemical bonds between dental ceramics and composite resins are critical for the success of repaired intraoral restoration [28,30]. Intraoral ceramic repair methods establish strong resin bonds after recommended surface treatments [30]. To minimize the influence of various conditions, the test specimens were fabricated as described by previous studies [15,18,28,30]. However, the direct extrapolation of these findings to the clinical performance of the restorative materials should be carefully made. 

The mechanical integrity and the bonding of the veneering ceramic to the framework material are the main parameters in the successful performance of veneer/framework restorations [31]. Sailer et al., in their research studies, have reported the clinical failure rate of the chipped ceramic layer to be 13% after three years and 15.2% after five years [32,33]. Numerous reports have made references to porcelain fracturing over the Zr core, with a rate of occurrence of 0.9–29.1% [34]. The delamination or fracture of the layered ceramic from the Zr framework can occur due to excessive shear stresses induced during continuous occlusal load applied during intraoral use, cyclic loading, or impact failure due to blow or accident [14]. This was also evident according to a report of clinical studies by Miura S et al. [34], in which they reported that the chipping of veneered ceramic occurred in the bicuspids and molars and did not occur in the front anterior teeth [34]. Mechanical problems such as occlusal loading, occlusion, frame/core design, and parafunctional habits could possibly be the elements responsible for inducing chipping [28]. The layered ceramic chipping can also be induced due to enormous tensile stresses developed in the veneering ceramic during their fabrication. These tensile stresses in the layered ceramic are developed because of the dissimilarity in coefficients of thermal expansion of the Zr core and overlying ceramic [35]. Due to the high strength of the Zr core and absence of creep deformation, there is a high risk of developing destructive stresses in the layering ceramic. Comparing it to metal-ceramic restoration, though the metal possesses high coefficient of thermal expansion than the ceramic, this difference is compensated by the creep associated with heating the metal component [36,37].

The ultimate tests for evaluating the bonding between different materials used for the restoration of teeth are clinical trials. However, laboratory tests are a viable option for gathering data quickly, easily, and on a specific parameter like bonding, while keeping the rest of the parameters constant [38]. The shear bond test that was employed in the current study is a reliable test for testing the repair bond strength of the repair materials to the Zr core and layering ceramic and has been used in similar research studies [15,18,28,30]. The reliability of this test was further confirmed by our findings, where the shear bond strength values of the standardized Zr core and layering ceramic specimens before the repair showed similar values (*p* = 0.062) for all the groups. In order to simulate the oral conditions, thermal cycling was performed for all the test specimens of the current study, since the aging of the specimens affects the bonding of the tested repair materials to the Zr core [39,40].

In the present study, significant differences (*p* = 0.02) in the repair bonding strength of five test groups to the same Zr core layered with ceramic were found. Significant differences were also observed for the shear bond strength between the pre and post-repair samples. Thus, the null hypothesis of similar values of shear bond strength after the repair of all five repair methods was rejected. These results are in line with previous studies by Blum et al. [41], Han et al. [28], Cristoforides et al. [39], and Kim et al. [42], who reported variations in the repaired shear bond strength of different repair systems investigated in their research studies. Another relevant and significant observation in agreement with the current study was the difference between the bond strength of three complete repair kits (Groups A, B, and C) and the two bonding materials (Groups D and E). The average difference between the two methods of repair was 2.73 MPa, which is significant keeping in view the overall average bond strength of 12.11 MPa for all the groups. This finding was also in line with Lee et al. [15], who used the complete repair kits for the repair of the fractured layered Zr crowns. For increasing the bonding strength between the Zr core and the repairing composite materials, the use of Zr primers is recommended in the literature [43]. This phenomenon was also observed in the current study, where the primers specifically designed for Zr were used for Group-C test specimens. The bond strength values of 13.79 + 1.32 MPa were noted for the Group-C (Signum, Zr repair kit) specimens, which was highest among all the tested groups. The Z-Primer (Group-C) contains organophosphate and carboxylic acid monomers which can add to the bond strength between the two materials due to chemical bond formation [44]. The organo-phosphates monomers have a phosphoric acid group that can bond to Zr surface oxides and a methacrylate group to copolymerize with organic monomers of the composite [43,44,45].

The modes of failure of the layered ceramic or repaired composite over the Zr core can be adhesive, cohesive, or mixed. The SEM examination of the specimens in the current study showed adhesive failure to be the common mode of failure for the specimens. This finding is in line with Kocaagaoglu et al.’s [46] research findings on ceramic repairs, who reported adhesive failure (84%) to be the major mode of failure for all the tested groups. These results could be related to the deboning of the repaired composite to the cores using vertical wedging forces, typical of shear bond testing.

The main limitation of this in vitro study was using square shapes specimens; a more realistic approach for testing the mechanical properties of the layered Zr would have been the fabrication of the test specimens in the form of crowns, for simulating the clinical scenario. The specimens of the current study were tested under constant vertical load applied until failure. The restorations intraorally are subjected to continuous fluctuations of temperature and moisture which affects their mechanical behavior. Future studies with long-term storage and repeated fatigue loadings under thermal conditions similar to the oral cavity for testing the long-term bond strength of layered Zr repair methods would be imperative. Though it is unlikely to have an in vitro test that accurately simulates and predicts the clinical scenario. Due to the above limitations, the results from this study were carried out under a controlled laboratory environment and should be interpreted with caution.

## 5. Conclusions

The repair methods tested showed variations in terms of their respective shear bond strength. The post-repair shear bond strength of all the tested groups was significantly lower (*p* < 0.05) than pre-repair ceramic to zirconia bonding. Among the tested groups, complete ceramic/zirconia repair systems showed better bonding between the repaired composite and zirconia core. The mean shear bond strength values for the repaired fractured layered zirconia showed an acceptable outcome in this in vitro research model however clinically may remain unpredictable due to the intraoral variables. The complete ceramic/zirconia repair systems can be the method of choice for repairing layered zirconia restorations. 

## Figures and Tables

**Figure 1 polymers-13-00910-f001:**
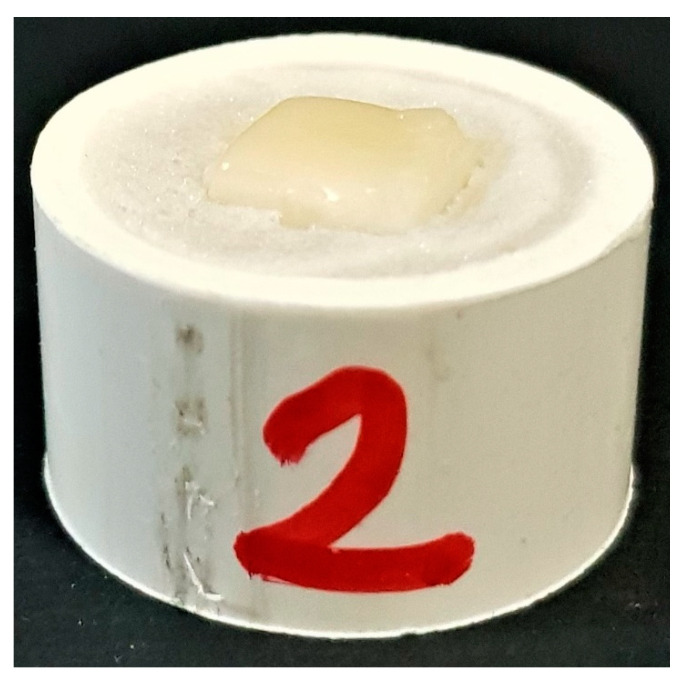
Zirconia specimen layered with ceramic and embedded in a resin block.

**Figure 2 polymers-13-00910-f002:**
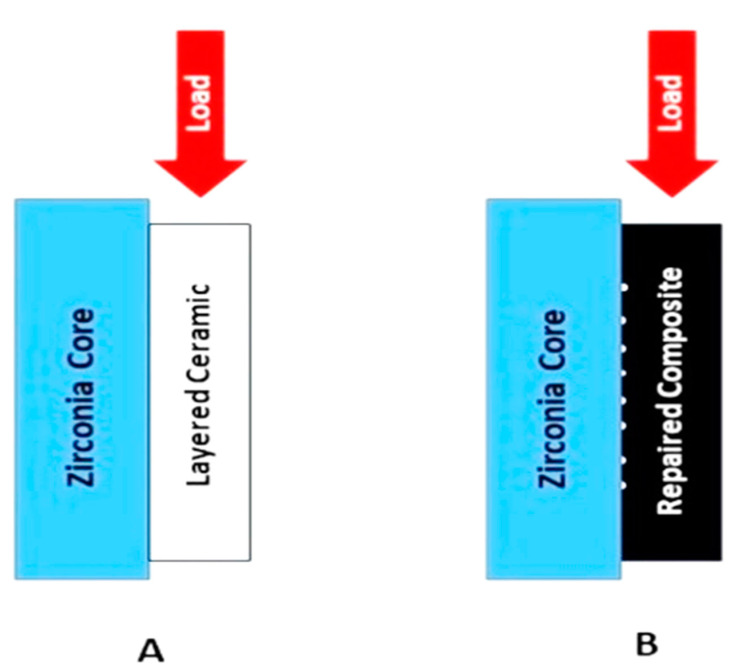
Diagrammatic representation of shear bond testing in universal testing machine; (**A**) zirconia core with ceramic; (**B**) zirconia core with repaired composite with some remnants of ceramic over the zirconia core.

**Figure 3 polymers-13-00910-f003:**
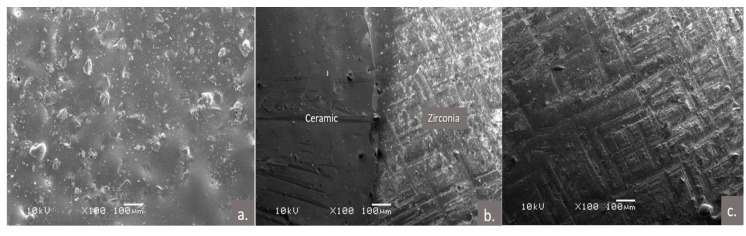
Representative scanning electron microscopic images (SEM) of; (**a**) ceramic surface; (**b**) ceramic zirconia interface; (**c**) zirconia surface after the ceramic chip off, small remnants of ceramic can be noted.

**Figure 4 polymers-13-00910-f004:**
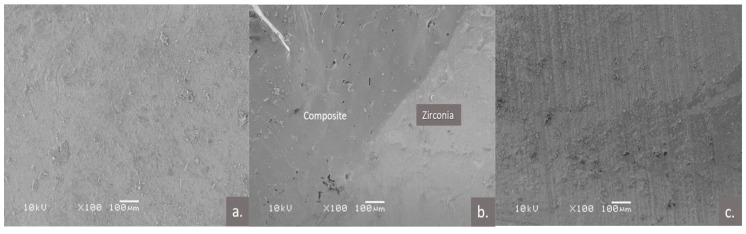
Representative scanning electron microscopic images of; (**a**) composite surface; (**b**) composite zirconia interface; (**c**) zirconia surface after the repaired composite chip off, small remnants of the composite can be noted.

**Table 1 polymers-13-00910-t001:** Details of the materials used in this research.

S. No.	Groups	Material	Trade Name	Manufacturer	Lot Number
1.	Zr	Zirconia	ZirCAD	Ivoclar Vivadent AG, Schaan/Liechhtenstein	X24851
2.	Cr	Ceramic	Ceram IPS e.max	Ivoclar Vivadent AG, Schaan/Liechhtenstein	X32767
3.	Group-A	Z-PRIME^TM^ Plus, Porcelain primer, Porcelain bonding resin	Intra Oral repair Kit	Bisco, INC. 1100W. Irving Park Rd. Schauburg, U.S.A.	81026A
4.	Group-B	Tetric Evoceram Light curing nano-hybrid composite, Monobond Plus, Heliobond	Ceramic repair N	Ivoclar Vivadent AG, Schaan/Liechhtenstein	90429A
5.	Group-C	Ceramic bond I, Ceramic bond II	Signum	Kulzer GmbH, Hanau, Germany	X51289
6.	Group-D	Scotch bond^TM^	Scotch bond^TM^	3M, St. Paul, MN	K654
7.	Group-E	Single bond^TM^	Single bond^TM^	3M, St. Paul, MN	1900004131

**Table 2 polymers-13-00910-t002:** Details of the repair procedures used for each of the five test groups.

Groups	Application Procedure
Group-A	Isolation of area to be repaired. Removal of glaze and bevel (45°) porcelain around the area to be repaired. Application of PORCELAIN ETCHANT (9.5% HF) for 90 s. Application of 1 coat of PORCELAIN PRIMER to the etched porcelain surface and allowing to dwell for 30 s. Air drying. Application 1 coat of Z-PRIME Plus to the exposed metal/zirconia/alumina and drying with an air syringe for 3–5 s. Application of a thin layer of PORCELAIN BONDING RESIN to the repair site. Spreading of composite evenly over the surface and light-curing. Completion of repair using composite and finishing/polishing.
Group-B	Isolation. Preparation of the defective ceramic surface. Application of Monobond N. and allowing to react for 60 s. Application of Heliobond, and light-curing for 10 s. Completion of the repair using composite and finishing/polishing.
Group-C	Preparation of the ceramic surface. Drying the surface thoroughly using oil-free air. Application of a thin layer of Signum ceramic bond I to the dust-free ceramic surface with a new brush or microbrush and allowing it to dry for about 10 s. Application of thin layer of Signum ceramic bond II and rub in for approx. 20–30 s—no light curing required! Composite build-up: Layer thickness <2 mm, light curing.
Group-D	Deglazing of the surface to be repaired. Application of the adhesive to the prepared tooth and rubbing it in for 20 s. Gently air-drying the adhesive for approximately 5 s to evaporate the solvent. Light curing for 10 s. Completion of repair using composite and finishing/polishing.
Group-E	Deglazing of the surface to be repaired. Application of the adhesive to the prepared tooth and rubbing it in for 20 s. Air drying the adhesive for about 5 s to evaporate the solvent. Light curing for 10 s. Completion of repair using composite and finishing/polishing.

**Table 3 polymers-13-00910-t003:** Descriptive statistics with mean, standard deviation, and ANOVA results of shear bond strength before and after the ceramic repair (N = 150).

	Material Groups	N	* Mean	Std. Deviation	95% Confidence Interval for Mean		Minimum	Maximum	ANOVA *p*-Value
					Lower Bound	Upper Bound			
Pre Repair Strength	Group-A	15	30.06	1.27	29.36	30.77	26.89	30.88	0.062
	Group-B	15	28.05	2.54	26.65	29.46	23.28	30.87	
	Group-C	15	27.66	2.35	26.35	28.96	22.86	30.86	
	Group-D	15	28.75	2.75	27.23	30.28	22.91	30.87	
	Group-E	15	27.87	3.02	26.20	29.55	22.82	30.88	
	Total	75	28.48	2.55	27.89	29.07	22.82	30.88	
Post Repair Strength	Group-A	15	13.44	3.04	11.75	15.12	8.29	20.70	0.002
	Group-B	15	12.37	2.49	10.99	13.75	7.22	16.34	
	Group-C	15	13.79	1.32	13.06	14.53	12.34	16.93	
	Group-D	15	9.77	4.77	7.13	12.41	4.30	23.90	
	Group-E	15	11.16	1.90	10.11	12.22	9.03	15.50	
	Total	75	12.11	3.24	11.36	12.85	4.30	23.90	

* Mean shear bond strength was calculated in mega Pascal’s (MPa).

**Table 4 polymers-13-00910-t004:** Pairwise comparison of shear bond strength in megapascals (MPa) of test groups before and after the ceramic repair by Paired Samples T-test.

Materials Tested	Shear Bond Strength	Mean	Std. Deviation	Std. Error Mean	Mean Difference	* *p* Value
Pair 1 Group-A	Pre-repair (n = 15)	30.06	1.27	0.32	16.62	0.000
	Post-repair (n = 15)	13.44	3.04	0.78		
Pair 2 Group-B	Pre-repair (n = 15)	28.05	2.54	0.65	15.68	0.000
	Post-repair (n = 15)	12.37	2.49	0.64		
Pair 3 Group-C	Pre-repair (n = 15)	27.66	2.35	0.60	13.86	0.000
	Post-repair (n = 15)	13.79	1.32	0.34		
Pair 4 Group-D	Pre-repair (n = 15)	28.75	2.75	0.71	18.98	0.000
	Post-repair (n = 15)	9.77	4.77	1.23		
Pair 5 Group-E	Pre-repair (n = 15)	27.87	3.02	0.78	16.71	0.000
	Post-repair (n = 15)	11.16	1.90	0.49		

* The comparison is significant at the *p*
< 0.05 level.

**Table 5 polymers-13-00910-t005:** Multiple comparisons and mean differences of the shear bond strength (MPa) after the ceramic repair by Post Hoc Tukey HSD test.

Dependent Variable	Groups	Comparison	Mean Difference	* Sig.
Post-Repair Shear Bond Strength0	A	B	1.066	0.860
		C	−0.357	0.997
		D	3.666 *	0.010
		E	2.273	0.229
	B	A	−1.066	0.860
		C	−1.423	0.680
		D	2.599	0.125
		E	1.2063	0.797
	C	A	0.357	0.997
		B	1.423	0.680
		D	4.023 *	0.003
		E	2.63	0.118
	D	A	−3.666 *	0.010
		B	−2.599	0.125
		C	−4.023 *	0.003
		E	−1.393	0.698
	E	A	−2.273	0.229
		B	−1.206	0.797
		C	−2.63	0.118
		D	1.393	0.698

* The mean difference was significant at the *p*
< 0.05 level.

## Data Availability

Data sharing not applicable.

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
