# Peer review of "Shear Bond Strength of Veneered Zirconia Repaired Using Various Methods and Adhesive Systems: A Comparative Study"

_polymers, 2021, doi:10.3390/polym13060910_

Round 1

Reviewer 1 Report

Dear authors,

This work is very good, several experiments were well done and some interesting results were obtained. The work is well written.

However, my main concern is that I don't think the work show much relevance with polymers. 

Author Response

Reviewers’ Comments: This work is very good, several experiments were well done and some interesting results were obtained. The work is well written.

However, my main concern is that I don't think the work show much relevance with polymers.

Authors’ response: Authors would like to thank you for reviewing and providing the valuable comments on this manuscript. The constructive feedback and affirmation of our work has greatly inspired us.

The present study investigated the shear bond strength of dental adhesive systems for a specific dental application and fits well within the scope of “Polymers”. Therefore, authors respectfully disagree with the reviewer’s last point (not much relevance with polymers) based on the following grounds:

  1. All the dental adhesives (included the one used in the present study) are composed of polymers (mainly polymethacrylates) and have been exhaustively investigated for its polymeric chemistry and properties in polymers related and interdisciplinary journals [Please see the quick search link; further detail can be provided on request];

https://scholar.google.co.uk/scholar?as_ylo=2017&q=dental+adhesive+polymer&hl=en&as_sdt=0,5 

  1. Considering the relevance of dental adhesives to polymers, a number of articles are published in the “polymers” journal [1] as well as in the specific special issue [2-5].
  2. After submission, the manuscript passed the initial evaluation for the scope and editorial checks and sent for peer-reviewing indicating its relevance and within the scope of the polymers.
  3. In addition, all the other three reviewers considered this manuscript within the scope of “polymers” (did not mention any issues related to the relevance); and critically reviewed the manuscript and provided the constructive feedback.
  4. A quick search showing recently published articles on dental adhesives in the Polymers can be reached by the following link;

https://www.mdpi.com/search?q=dental+adhesive&journal=polymers

Based on the above points, it is evident that this manuscript fits well with in the scope of the journal and the special issue. We would be grateful if the worthy reviewer and academic editor kindly consider our response/rebuttal acceptable.

Reference

  1. Sarikaya, R.; Song, L.; Ye, Q.; Misra, A.; Tamerler, C.; Spencer, P. Evolution of Network Structure and Mechanical Properties in Autonomous-Strengthening Dental Adhesive. Polymers 2020, 12, 2076. https://doi.org/10.3390/polym12092076
  2. Khan, A.A.; Perea-Lowery, L.; Al-Khureif, A.A.; AlMufareh, N.A.; Eldwakhly, E.; Säilynoja, E.; Vallittu, P.K. Interfacial Adhesion of a Semi-Interpenetrating Polymer Network-Based Fiber-Reinforced Composite with a High and Low-Gradient Poly(methyl methacrylate) Resin Surface. Polymers202113, 352. https://doi.org/10.3390/polym13030352
  3. AlFawaz, Y.F.; Almutairi, B.; Kattan, H.F; Zafar, M.S.; Farooq, I.; Naseem, M.; Vohra, F.; Abduljabbar, T. Dentin Bond Integrity of Hydroxyapatite Containing Resin Adhesive Enhanced with Graphene Oxide Nano-Particles—An SEM, EDX, Micro-Raman, and Microtensile Bond Strength Study. Polymers202012, 2978. https://doi.org/10.3390/polym12122978
  4. Al-Hamdan, R.S; Almutairi, B.; Kattan, H.F; Alsuwailem, N.A.; Farooq, I.; Vohra, F.; Abduljabbar, T. Influence of Hydroxyapatite Nanospheres in Dentin Adhesive on the Dentin Bond Integrity and Degree of Conversion: A Scanning Electron Microscopy (SEM), Raman, Fourier Transform-Infrared (FTIR), and Microtensile Study. Polymers202012, 2948. https://doi.org/10.3390/polym12122948
  5. Khan, A.S.; Ur Rehman, S.; AlMaimouni, Y.K.; Ahmad, S.; Khan, M.; Ashiq, M. Bibliometric Analysis of Literature Published on Antibacterial Dental Adhesive from 1996–2020. Polymers202012, 2848. https://doi.org/10.3390/polym12122848

Reviewer 2 Report

Syed Rashid Habid, et al. investigated shear bond strength of veneered zirconia repaired using various methods and adhesive systems: a comparative study. Basically, this is a good study. I just have some suggestions for a minor revision.

For Fig. 3 and Fig. 4, please state whether these figures are representative for repeated experiments. How many times have been repeated?

Author Response

Reviewers’ Comments: Syed Rashid Habid, et al. investigated shear bond strength of veneered zirconia repaired using various methods and adhesive systems: a comparative study. Basically, this is a good study. I just have some suggestions for a minor revision.

Authors’ response

Dear Reviewer,

Authors would like to thank you for reviewing and providing the valuable comments on this manuscript. The constructive feedback and affirmation of our work has greatly inspired us. Based on the suggestions, authors thoroughly revised the manuscript. Please find a point-by-point response to the reviewer’s comments below:

Reviewers’ Comments: For Fig. 3 and Fig. 4, please state whether these figures are representative for repeated experiments. How many times have been repeated?

Authors’ response: Thank you very much for insightful comment. Although we scanned all the specimens by SEM, it is not feasible and impactful to include all SEM images. Authors confirm that all the images presented in the Fig. 3 and Fig 4 are representative images to compare the characteristic fractographic differences of various failed specimen. This information has been added to the figure captains. 

Reviewer 3 Report

The manuscript entitled “ Shear bond strength of veneered zirconia repaired using various methods and adhesive systems: a comparative study” is not an original research because similar research was published recently “Maarof, Mohamed, Mosaad Elgabarouny, Amr Shebl, and Rania Badawy. "Shear Bond Strength of Composite Repair Material to Ceramo-metallic and zirconium Restorations Using A New Adhesive System." Dental Science Updates 1, no. 1 (2020): 51-62.; https://dsu.journals.ekb.eg/article_74964.html”

I have also some other comments for authors:

Please mention the analytical techniques employed in abstract.

Figure 2 is blurry, please revise.

How did the authors prepare the samples for SEM analysis, please describe especially figure 3b.

Figures 3 and 4 are important; please redraw in a horizontal manner.

Conclusion is just a repetition of abstract, please describe the concluded significant data and results.

Author Response

Reviewers’ Comments: The manuscript entitled “Shear bond strength of veneered zirconia repaired using various methods and adhesive systems: a comparative study” is not an original research because similar research was published recently “Maarof, Mohamed, Mosaad Elgabarouny, Amr Shebl, and Rania Badawy. "Shear Bond Strength of Composite Repair Material to Ceramo-metallic and zirconium Restorations Using A New Adhesive System." Dental Science Updates 1, no. 1 (2020): 51-62.; https://dsu.journals.ekb.eg/article_74964.html”

Authors’ response

Dear Reviewer,

Authors would like to thank you for reviewing and providing the valuable comments on this manuscript. Based on the suggestions, authors thoroughly revised the manuscript and responded to the reviewer’s comments.

Authors are very much thankful to the reviewer for bringing this article in our notice. First, we are sorry to miss this article during our literature search probably due to the fact this it is not indexed by any electronic database (such as SCOPUS, PubMed, Web of Science etc). Although the article by Maroof et al has a similar title, it presents a totally different work and authors have presented the key differences for further clarification (Table below).

Maroof et al

Present study

Aim

Evaluate shear bond strength (SBS) of composite repair material to ceramo-metalic and zirconium restorations using one adhesive system.

A comprehensive study design to compare the shear bond strength of different methods and various adhesive systems for repairing zirconia (Zr) cores layered with feldspathic porcelain.

Materials

Investigated only one type of bonding agent.

Five different repair materials or methods used only for Zirconia repair were tested and compared.

Methods

Only tested the bond strength

Repair of the Zirconia specimens after the fracture of the layered ceramic and then aging (thermocycling) of the ceramics. To simulate clinical scenario.

The methodology employed is unique, where the pre and post comparison of shear bond strength before and after ceramic repair with five different repair systems are also carried out to compare their functionality.

This has not been investigated/published previously with any research. Most of the studies have only tested the shear bond strength of the composite repair to the Zirconia with no Zirconia layered ceramic fracture.

Fractography

None

Detailed fractography of the deboned surfaces is presented using SEM; that has provided useful and impactful data present in this study.

As presented above in the table, it is evident that both the studies are clearly different in several aspects including “aim, materials used, methodology, study design, results and their potential impact. Therefore, the authors confirm that this manuscript presents an original research and will contribute a sufficient novelty to the scientific literature and the audience. At present, the authors would not go into further details of the experimental design and write-up of the published research. However, we are open to provide further details and happy to respond to any further queries regarding this point.

Reviewers’ Comments: I have also some other comments for authors:

Please mention the analytical techniques employed in abstract.

Authors’ response: Thank you for the valuable feedback. As suggested, authors have included the details of the analytical techniques in the abstract (Lines 21-27).

Reviewers’ Comments: Figure 2 is blurry, please revise.

Authors’ response: Figure 2 has been replaced by a higher resolution image (Figure 2).

Reviewers’ Comments: How did the authors prepare the samples for SEM analysis, please describe especially figure 3b.

Authors’ response: Authors have added the details of preparing specimens for SEM analysis (Line 219-224).

Reviewers’ Comments: Figures 3 and 4 are important; please redraw in a horizontal manner.

Authors’ response: Thank you for the suggestion, the figures 3 and 4 are redrawn horizontally as suggested (Figure 3 and 4).

Reviewers’ Comments: Conclusion is just a repetition of abstract, please describe the concluded significant data and results.

Authors’ response: Authors are very much thankful for the insightful feedback. Authors have revised the conclusion section and represented the key findings and results (Line 364-372).

Reviewer 4 Report

Dear Editor,

The manuscript with title "Shear bond strength of veneered zirconia repaired using various methods and adhesive systems: a comparative study" is very interesting paper which deals with methodology, results and application of the results in the field of dental medicine. The results from the article are clearly discussed and description of the methodology is acceptable. I have few suggestions regarding minor correction of the manuscript:

  1. Results from one-way ANOVA (Table 3) should be clearly presented (in terms of statistical significance between each each material groups (A, B, C, D and E)
  2. Results from one-way ANOVA (Table 4), last column p value for all 4 groups is 0.000 which means there is no statistical difference between the groups (this can be discussed why)
  3. Line 381-382: The authors should explained in the conclusion why they stated that "The mean shear bond strength values for the repaired fractured layered zirconia showed an acceptable outcome" but why  unpredictable? 

Author Response

Reviewers’ Comments: The manuscript with title "Shear bond strength of veneered zirconia repaired using various methods and adhesive systems: a comparative study" is very interesting paper which deals with methodology, results and application of the results in the field of dental medicine. The results from the article are clearly discussed and description of the methodology is acceptable. I have few suggestions regarding minor correction of the manuscript:

Authors’ response

Dear Reviewer,

Authors would like to thank you for reviewing and providing the valuable comments on this manuscript. The constructive feedback and affirmation of our work has greatly inspired us. Based on the suggestions, authors thoroughly revised the manuscript. Please find a point-by-point response to the reviewer’s comments below:

Reviewers’ Comments: 1. Results from one-way ANOVA (Table 3) should be clearly presented (in terms of statistical significance between each each material groups (A, B, C, D and E)

Authors’ response: Thank you very much for the insightful comment. All the data and statistical analysis were performed using the SPSS software and presented the overall comparison by ANOVA (Table 3). Accordingly, for one to one comparison between the groups, we applied the Post Hoc Tukey’s test and presented the findings separately (Table 5; Page 8-9).

Reviewers’ Comments: 2. Results from one-way ANOVA (Table 4), last column p value for all 4 groups is 0.000 which means there is no statistical difference between the groups (this can be discussed why)

Authors’ response: For the statistical analysis, the P value (P<0.05) was considered statistically significant as mentioned in the footnotes (Table 4). According to the data analysis, the P value for all the groups (0.000) was less than 0.05 (P<0.05) suggesting a statistically significant difference while comparing the pre and post repair, shear bond strength, in each group. For clarity, this information has been added to the Table 5 (Lines Footnotes).

Reviewers’ Comments: 3. Line 381-382: The authors should explained in the conclusion why they stated that "The mean shear bond strength values for the repaired fractured layered zirconia showed an acceptable outcome" but why unpredictable?

Authors’ response: Authors are very much thankful to for the valuable comments. As suggested, the mentioned statement has been corrected and modified for further clarity For the statistical analysis, the P value (P<0.05) was considered (Line 365-370).

Round 2

Reviewer 3 Report

none